# Antenatal *Ureaplasma* Infection Causes Colonic Mucus Barrier Defects: Implications for Intestinal Pathologies

**DOI:** 10.3390/ijms25074000

**Published:** 2024-04-03

**Authors:** Charlotte van Gorp, Ilse H. de Lange, Matthias C. Hütten, Carmen López-Iglesias, Kimberly R. I. Massy, Lilian Kessels, Kèvin Knoops, Iris Cuijpers, Mireille M. J. P. E. Sthijns, Freddy J. Troost, Wim G. van Gemert, Owen B. Spiller, George M. H. Birchenough, Luc J. I. Zimmermann, Tim G. A. M. Wolfs

**Affiliations:** 1Department of Pediatrics, School for Oncology and Reproduction (GROW), Maastricht University, 6229 ER Maastricht, The Netherlands; c.vangorp@maastrichtuniversity.nl (C.v.G.); matthias.hutten@mumc.nl (M.C.H.); k.massy@maastrichtuniversity.nl (K.R.I.M.); lilian.kessels@maastrichtuniversity.nl (L.K.); luc.zimmermann@mumc.nl (L.J.I.Z.); 2Department of Pediatrics, School of Nutrition and Translational Research in Metabolism (NUTRIM), Maastricht University, 6229 ER Maastricht, The Netherlands; i.delange@maastrichtuniversity.nl; 3Neonatology, Department of Pediatrics, University Hospital RWTH Aachen, 52074 Aachen, Germany; 4Microscopy CORE Lab, Maastricht Multimodal Molecular Imaging Institute (M4I), Maastricht University, 6211 LK Maastricht, The Netherlands; c.lopeziglesias@maastrichtuniversity.nl (C.L.-I.); k.knoops@maastrichtuniversity.nl (K.K.); 5Food Innovation and Health, Department of Human Biology, School of Nutrition and Translational Research in Metabolism (NUTRIM), Maastricht University, 5911 BV Venlo, The Netherlands; iris.cuijpers@maastrichtuniversity.nl (I.C.); mireille.sthijns@maastrichtuniversity.nl (M.M.J.P.E.S.); f.troost@maastrichtuniversity.nl (F.J.T.); 6Department of Surgery, School of Nutrition and Translational Research in Metabolism (NUTRIM), Maastricht University, 6229 ER Maastricht, The Netherlands; wim.van.gemert@mumc.nl; 7Division of Infection and Immunity, School of Medicine, Cardiff University, Cardiff CF14 4XW, UK; spillerb@cardiff.ac.uk; 8Department of Medical Biochemistry, Institute of Biomedicine, University of Gothenburg, 40530 Gothenburg, Sweden; george.birchenough@gu.se; 9Department of Biomedical Engineering (BMT), School for Cardiovascular Diseases (CARIM), Maastricht University, 6229 ER Maastricht, The Netherlands

**Keywords:** *Ureaplasma*, perinatal inflammation, intestinal mucus barrier, goblet cell, necrotizing enterocolitis, colon

## Abstract

Chorioamnionitis is a risk factor for necrotizing enterocolitis (NEC). *Ureaplasma parvum* (UP) is clinically the most isolated microorganism in chorioamnionitis, but its pathogenicity remains debated. Chorioamnionitis is associated with ileal barrier changes, but colonic barrier alterations, including those of the mucus barrier, remain under-investigated, despite their importance in NEC pathophysiology. Therefore, in this study, the hypothesis that antenatal UP exposure disturbs colonic mucus barrier integrity, thereby potentially contributing to NEC pathogenesis, was investigated. In an established ovine chorioamnionitis model, lambs were intra-amniotically exposed to UP or saline for 7 d from 122 to 129 d gestational age. Thereafter, colonic mucus layer thickness and functional integrity, underlying mechanisms, including endoplasmic reticulum (ER) stress and redox status, and cellular morphology by transmission electron microscopy were studied. The clinical significance of the experimental findings was verified by examining colon samples from NEC patients and controls. UP-exposed lambs have a thicker but dysfunctional colonic mucus layer in which bacteria-sized beads reach the intestinal epithelium, indicating undesired bacterial contact with the epithelium. This is paralleled by disturbed goblet cell MUC2 folding, pro-apoptotic ER stress and signs of mitochondrial dysfunction in the colonic epithelium. Importantly, the colonic epithelium from human NEC patients showed comparable mitochondrial aberrations, indicating that NEC-associated intestinal barrier injury already occurs during chorioamnionitis. This study underlines the pathogenic potential of UP during pregnancy; it demonstrates that antenatal UP infection leads to severe colonic mucus barrier deficits, providing a mechanistic link between antenatal infections and postnatal NEC development.

## 1. Introduction

One of the main risk factors for premature birth, defined as birth before 37 weeks of gestation, is chorioamnionitis [1]. The prevalence of chorioamnionitis is inversely correlated to gestational age (GA) at birth, with a prevalence of ~40% in infants born at a GA of 31–33 weeks following spontaneous preterm labor [2]. Chorioamnionitis is an inflammatory reaction of the uterine membranes (chorion and amnion) and cells in the amniotic fluid, often caused by microbial invasion into the amniotic cavity [1,3]. It can lead to a subsequent systemic inflammatory response called fetal inflammatory response syndrome (FIRS) [4,5]. Chorioamnionitis and FIRS are associated with adverse outcomes and postnatal pathologies in preterm neonates, including intestinal emergency necrotizing enterocolitis (NEC) [6]. NEC is characterized by gastrointestinal mucosal inflammation that can progress to tissue necrosis and is related to high mortality rates (20–30%) and high morbidity rates in survivors [7]. NEC most frequently affects the terminal ileum but also regularly involves the (proximal) colon [8,9]. Most in vivo experimental studies investigating NEC pathophysiology focus on the small intestine [10]. Colonic barrier changes associated with NEC development, particularly those of the important mucus barrier, remain under-investigated.

*Ureaplasma* species, most notably *Ureaplasma parvum* (UP), are the most frequently cultured microorganisms from the amniotic fluid and fetal membranes, with evidence of chorioamnionitis. Although their pathogenicity is still debated [3], intra-amniotic (IA) infection with genital mycoplasmas (including ureaplasmas) is associated with increased IA inflammation compared to other microorganisms in preterm premature rupture of membranes [11]. In addition, we previously observed increased fetal systemic inflammation (plasma IL6 levels), low grade intestinal inflammation and structural enteric nervous system alterations following IA UP exposure in earlier studies in an ovine chorioamnionitis model [12,13], indicating that despite the absence of acute severe intestinal damage following IA UP exposure, it is far from harmless. In the current study, we hypothesized that antenatal exposure to UP elicits mucus barrier damage to the premature colon, potentially causing a predisposition toward postnatal NEC development.

In the complex pathophysiology of NEC, intestinal mucosal barrier disruption is of growing interest [7]. In a healthy colon, a densely packed inner mucus layer (IML) prevents bacteria from reaching the intestinal epithelium [14,15]. In addition, a less cohesive outer mucus layer exists in which commensal intestinal microbiota live in symbiosis with the host [16,17]. Goblet cells produce mucin-2 (MUC2), the main component of the mucus barrier in the intestinal tract [16,18]. In addition, colonocytes have a key role in regulating the potential interaction with invading microorganisms [18,19]. During NEC, the intestinal mucosal barrier is disrupted, characterized by increased epithelial permeability, reduced goblet cell numbers and decreased mucin secretion [20,21,22]. In case of loss of functional integrity of the mucus layer, bacteria can reach the epithelium and subsequently provoke adverse outcomes [19]. It is known that inflammatory bowel diseases (IBD), especially ulcerative colitis, are characterized by a permeable colonic mucus layer [23,24,25]. Moreover, intestinal permeability is increased in premature neonates, and this permeability is adversely related with infant age [26,27,28], making the premature intestine more vulnerable to the development of intestinal disorders characterized by barrier dysfunction such as NEC [29,30].

Pathophysiological mechanisms implicated in NEC and other inflammatory conditions, such as IBD, that are considered to contribute to gut (mucus) barrier dysfunction include endoplasmic reticulum (ER) stress [31,32,33,34,35] and oxidative stress [35,36,37]. Under physiological circumstances, the ER enables proper protein folding, modification and secretion [38,39]. ER protein folding is an oxidative process that produces ROS that is normally adequately scavenged by molecules of the cellular redox systems, including reduced glutathione (GSH) [40]. An important transcription factor in this context is Nuclear factor erythroid 2-related factor 2 (Nrf2). Nrf2 translocation leads to the upregulation of several genes involved in the endogenous antioxidant defense response [41]. In the case of ER stress, unfolded proteins accumulate, which leads to the synthesis and secretion of dysfunctional proteins [39] including mucins [42,43]. ER stress is regulated by an unfolded protein response (UPR) in which the central regulatory protein binding immunoglobulin protein (BiP) can induce several molecular cascades to resolve unfolded proteins [44,45]. Prolonged unresolved ER stress becomes pathological, leading to the upregulation of C/EBP homologous protein (CHOP), a promotor of epithelial cell death [46]. This pathogenic process is one of the mechanisms that represents NEC pathology [31,47]. ER stress and oxidative stress are closely intertwined [48]. GSH plays an important role in preventing UPR, as it is used for S-glutathionylation of ER proteins, such as BiP and protein disulfide isomerase (PDI) [49]. A decrease in S-glutathionylated of these ER proteins is associated with ER stress [49]. On the other hand, ER stress causes excessive production of ROS [38,48].

To conclude, in the current study, we investigated whether colonic mucus barrier changes already occur during pregnancy following IA UP infection to obtain insight into the effects of antenatal UP exposure on the fetal colon and to provide mechanistic explanations for the well-known association between chorioamnionitis and NEC. For this purpose, we used an established pre-clinical ovine chorioamnionitis model in which we studied colonic mucus layer thickness and functional integrity and the expression of several goblet cell markers. Furthermore, the underlying mechanisms, including ER stress and redox status, and cellular morphological alterations were studied in the proximal colon by performing transmission electron microscopy (TEM), among others. Lastly, our experimental findings were verified in clinical colonic samples from NEC patients and controls to validate the clinical significance of our experimental data.

## 2. Results

### 2.1. Intra-Amniotic UP Exposure Causes a Thicker but Dysfunctional Mucus Layer in the Premature Ovine Colon

To study the colonic mucosal barrier, total mucus layer (TML) thickness (assessed with 10 µm beads) and functional mucus barrier integrity (thickness of the inner part of the mucus layer, assessed with 1 µm beads) were evaluated (Figure 1). Unexpectedly, the TML was significantly thicker (*p* < 0.05) in UP-exposed animals compared to controls (Figure 1A). A functional inner part of the mucus layer was observed in premature control lambs, as bacteria-sized 1 µm beads did not reach the colonic epithelium (Figure 1B,C). Lambs that were IA exposed to UP had a compromised functional integrity of their mucus barrier with a reduced bead-epithelium distance (*p* < 0.05; Figure 1B,D) and numerous beads reaching the epithelial surface. In addition, colonic epithelial organization was disrupted in the IA UP-exposed animals compared to the controls (Figure 1C,D). 

### 2.2. Alterations in Goblet Cell Maturation and Differentiation in the Premature Ovine Colon Following Intra-Amniotic UP Exposure

Since we observed changes in mucus layer thickness, mucus barrier functional integrity and colonic epithelial organization following IA UP exposure, we evaluated whether goblet cell numbers, maturation and differentiation were changed in the course of UP-induced chorioamnionitis. Goblet cell numbers were evaluated by measuring the number of colonic epithelial cells positively stained for MUC2, the main mucin product. UP exposure did not lead to altered MUC2+ goblet cell numbers compared to the preterm controls (Figure 2A,B,E). Since the (O-glycosylated) MUC2 product is formed from ‘immature’ (non-O-glycosylated) apomucin-2 (iMUC2) in colonic goblet cells, the number of iMUC+ cells was also analyzed. Compared to premature control lambs (Figure 2C,F), we detected significantly more iMUC2+ goblet cells (*p* < 0.05) following IA UP exposure (Figure 2D,F). More intense iMUC2 staining was observed at the lower crypts of the control lambs compared to the upper crypt (Figure 2C). Interestingly, this localization-dependent difference in iMUC2 staining intensity was not observed in 7 d IA UP-exposed lambs, in which the upper and lower crypt iMUC2 staining intensity was comparable (Figure 2D). 

The transcription factor Sterile Alpha Motif Pointed Domain-containing Ets Transcription Factor (SPDEF) is responsible for the terminal differentiation and maturation of goblet cells [50]. The number of SPDEF+ goblet cells was unaltered following 7 d of IA UP exposure compared to controls (Figure 3A–C). Collectively, these data indicate that 7 d IA UP exposure increases iMUC expression, mainly in upper crypt goblet cells in the ovine premature proximal colon.

### 2.3. IA UP Exposure Induces Pro-Apoptotic ER Stress in the Proximal Colon of Preterm Lambs

We studied the association between the observed alterations in iMUC expression in upper crypt goblet cells and intestinal epithelial ER stress, since the ER is of pivotal importance for the protein folding of important secretory proteins, including MUC2. We analyzed the protein expression levels of BiP, a chaperone protein that is involved in resolving unfolded protein accumulation [44,45], and the number of cells positively stained for CHOP, a driver of pro-apoptotic ER stress [21], in the proximal colon of preterm lambs.

The protein expression level of BiP was higher in the upper crypt than in the lower crypt in both groups (Figure 4A,B), which corresponds with the higher baseline mucus secretion of goblet cells in the upper crypt and its related increased protein folding demand when compared to goblet cells in the lower crypt. After 7 d IA UP exposure, the expression level of BiP was not changed compared to controls (Figure 4B). In contrast, 7 d IA UP exposure significantly increased the number of CHOP+ cells (*p* < 0.05) compared to control lambs (Figure 4C–E), indicating that UP-induced chorioamnionitis is associated with pro-apoptotic ER stress in the premature proximal colon.

### 2.4. Redox State and Oxidative Stress Level in the Proximal Preterm Colon Following Intra-Amniotic UP Exposure

Since ER stress is associated with mitochondrial dysfunction and induces ROS production [38], we investigated the redox state (GSH/GSSG ratio) and the mRNA expression of oxidative stress-related genes (GCLC, TXNRD1 and HMOX1) in the proximal colon of preterm lambs following IA UP exposure compared to controls (Figure 5). The GSH:GSSG ratio in the proximal colon was lower than the interquartile range of controls in 5 out of 8 animals in the IA UP-exposed group (Figure 5A), indicative of a biologically relevant effect, although this difference did not reach statistical significance. This could be explained by both small decreases in GSH protein expression (Figure 5C) and small increases in GSSG protein expression (Figure 5E) following IA UP exposure.

No differences were found in the gene expression levels of GCLC and TXNRD1 between the control and 7 d UP groups in the preterm ovine colon (Figure 5B,D). Of note, mRNA expression levels of HMOX1, an enzyme that has immunomodulatory capacities and prevents oxidative damage during stress [51,52], tended to be decreased (*p* = 0.06) in the proximal colon following IA UP exposure for 7 days compared to controls (Figure 5F).

### 2.5. Mucosal Barrier Cell Alterations on Cellular and Organelle Level Measured by TEM Imaging Following IA UP Exposure in the Premature Proximal Ovine Colon

Since ER stress and oxidative stress are associated with mitochondrial dysfunction [38] and a previous study in the same animals showed overt cellular morphological alterations of the ileum following IA UP exposure [53], we investigated morphological alterations at the cellular and organelle levels in the proximal preterm colon with TEM (Figure 6 and Figure 7).

In the upper crypts of IA UP-exposed lambs, a damaged colonic epithelium (both goblet cells and colonocytes) was detected, with edema between the epithelial cells, which was not observed in preterm control lambs (Figure 6A,B). IA UP exposure for 7 days led to an increased presence of glycogen in the lower crypts of the proximal colon, compared to preterm control lambs (Figure 6C,D).

Since goblet cells and colonocytes are responsible for the formation and regulation of the colonic mucosal barrier, we assessed these cells in more detail at the organelle level. In utero, UP exposure disturbed mitochondrial morphology in colonocytes along the crypt axis of the premature proximal colon (Figure 7). In the upper crypts of control preterm animals, the mitochondria of colonocytes consist of parallel-organized cristae perpendicular to the mitochondrial membranes (Figure 7A). In contrast, IA UP exposure leads to severe disruption of the mitochondrial morphology at the upper crypts; these mitochondria were globular shaped, the mitochondrial matrix were less electron dense, and the mitochondrial cristae were disrupted and less abundant compared to controls (Figure 7A,B). While the mitochondrial morphological appearance in the lower crypts was comparable to that in upper crypts for control lambs (Figure 7C), UP exposure disrupted mitochondrial morphology in colonocytes in the lower crypts, albeit to a lower extent than in the upper crypts. The mitochondrial cristae of colonocytes in the lower crypts of 7 d IA UP-exposed lambs appeared to have an abnormal structure characterized by a loss of parallel organization (i.e., not perpendicular relative to the mitochondrial membranes) and lobular shaped morphology (Figure 7D).

Overt alterations of mitochondrial appearance, glycogen accumulation and epithelial damage were found in two-thirds of the IA UP-exposed animals, which corresponds to the percentage of affected animals in an earlier study of the terminal ileum [53] and the clinical variable phenotype caused by UP infections [54]. Collectively, whereas mild mitochondrial alterations and glycogen accumulation are observed in the lower colonic crypts of UP-exposed animals, more pronounced and severe signs of mucosal cell injury and mitochondrial morphological disturbance are observed in the upper colonic crypts following IA UP exposure.

### 2.6. Mucosal Barrier Cell Alterations on Cellular and Organelle Level Measured by TEM Imaging along the Colonic Crypt Axis of NEC Patients

Since chorioamnionitis is a well-known risk factor for the development of NEC [6], the findings observed in IA UP-exposed ovine proximal colons were compared to colonic biopsies from NEC patients and controls using TEM (Figure 8 and Figure 9; based on TEM analysis moderately affected NEC patient, clinical Modified Bell’s Criteria IIIB) and Appendix A (based on TEM analysis severely affected NEC patient, Modified Bell’s Criteria IIIB).

The upper crypt of infants with NEC was severely injured; no identifiable mucosal barrier cells (goblet cells and colonocytes) were observed, in contrast to the nicely structured upper crypt containing goblet cells filled with mucin granules detected in controls (Figure 8A,B). The lower colonic crypt of NEC patients also had a disrupted cellular morphology, characterized by disrupted cell organization and decreased numbers of recognizable goblet cells compared to the controls (Figure 8C,D).

The mitochondrial appearance of colonocytes in the upper colonic crypts of NEC patients was disrupted. It is characterized by more globular-shaped mitochondria containing a less electron-dense matrix and disorganized and fewer identifiable cristae, compared to mitochondria with nicely organized cristae in the controls (Figure 9A,B). Similar but less severe mitochondrial morphological alterations were observed in the colonic lower crypt of NEC patients compared to controls (Figure 9C,D).

### 2.7. Detection of Ureaplasma parvum in the Proximal Colonic Epithelium of IA UP-Exposed Lambs

Based on our combined findings, we aimed to investigate whether the alterations found in the colonic crypts of premature lambs following IA UP exposure are associated with direct contact between the UP and the colonic epithelium. Therefore, we performed immunofluorescence staining to assess whether UP is present in the colonic epithelium of animal IA exposed to UP for 7 days. Interestingly, UP was present in the proximal colon of all UP-exposed lambs (Figure 10), whereas no positive signal was detected in the premature controls. UP was often localized around the nucleus of the colonic epithelial cells (Figure 10).

## 3. Discussion

In this experimental ovine study, we demonstrated that an infection of the amniotic fluid with UP, the most frequently isolated bacteria in pregnant women with chorioamnionitis, leads to hampered functional integrity of the IML in the premature colon. Moreover, we observed that intrauterine UP exposure is associated with ER stress and oxidative stress in the goblet cells and colonocytes of the premature colon, implying these as pathophysiological mechanisms involved in these mucus barrier changes. We could further demonstrate clinical significance, as our observations on a cellular level showed remarkable overlap with clinical NEC samples from human patients.

The reduced functional integrity of the IML following antenatal UP exposure correlated with a thicker overall mucus layer. Since the number of goblet cells was unaltered, this could have resulted from mucus hypersecretion by colonic goblet cells. To further unravel whether mucus hypersecretion solely reflects a failing mechanism to retain mucus barrier integrity or contributes to it through the induction of goblet cell dysfunction, we examined in detail the processes that could underlie the identified mucus barrier changes.

First, we provide evidence for the involvement of disrupted ER function due to unresolved ER stress. Following translation, MUC2 proteins undergo a complex process of protein folding and other post-translational modifications in which the ER plays a crucial role [14]. Abnormalities in this process have previously been associated with reduced mucus quantity and quality [55]. In our model, we observed pro-apoptotic ER stress of the colonic epithelium with an increased number of CHOP+ epithelial cells following IA UP exposure. In addition, the increased iMUC staining intensity in the upper crypts and increased numbers of iMUC+ cells following IA UP exposure are considered to result from iMUC protein accumulation due to unresolved ER stress. We previously observed pro-apoptotic ER stress also in the small intestine (terminal ileum) of IA UP-exposed lambs [53]. Interestingly, in contrast to our previous findings in the small intestine, we did not observe overt morphological changes in the ER of goblet cells in the colon. This could be related to the higher rates of spontaneous mucus excretion in the colon compared to the ileum [56], suggesting the colon might be better ‘equipped’ to manage higher mucus secretion rates. In addition, baseline ER stress levels are described to be higher in the ileum than in the colon for healthy human controls [57].

Second, mitochondrial dysfunction appears to play a role since we found signs of these pathophysiological mechanisms in the preterm colon following IA UP exposure. The lower, albeit not statistically significant, GSH:GSSG ratio in the current study suggests a lower redox status, which could indicate previous exposure to elevated oxidative stress levels. Of note, these findings are considered to be an underestimation of the changes at the epithelial level, as the measurements were performed on whole tissue homogenates, whereas most prominent effects are expected in the colonic epithelium—the site of direct UP contact and invasion. In line with this hypothesis, morphological changes of mitochondria associated with oxidative stress (i.e., less electron dense matrix and disorganization of the inner-membrane cristae) [58] were observed in the colonic epithelium of UP-exposed lambs, predominantly in the upper crypts. In addition, a complete cellular lysate was used for analysis, rather than specific lysates from the ER or mitochondrial cellular compartments [59]. Remarkably, mitochondrial changes associated with oxidative stress were mainly observed in colonocytes rather than in goblet cells. The observation in colonocytes could be related to UP-induced inflammation; in a large single-cell RNA sequencing study in the human colon of healthy controls and patients with ulcerative colitis, increased expression of oxidative stress pathway related genes including NOS2 was observed in crypt-top colonocytes during inflammation [60]. Whether oxidative stress in colonocytes ‘spills over’ to goblet cells or influences goblet cell function by other processes, such as altered inflammasome signaling [61] remains to be elucidated.

In the current study, remarkable co-localization was observed between (mild) disrupted mitochondria in the lower crypts and intracellular glycogen accumulation. Accumulation of glycogen in the intestinal epithelium has previously been described in early human fetal development, where it may serve as storage for metabolic needs or could be a source for the later production of glycoproteins such as mucins [62]. It was also observed in fetal lambs following intra-uterine growth restriction (IUGR), which was considered to be the result of developmental obstruction [63]. From these findings, it is tempting to speculate that glycogen accumulation in the current study may represent a more ‘immature’ state of the intestinal epithelium or changes in cell metabolism following UP exposure in utero.

Importantly, the observed presence of intracellular UP in the proximal colon 7 d after induction of IA infection indicates that the disrupted functional integrity of the IML and related cellular processes result from direct contact between the UP and the colonic epithelium. In line with our findings, in an in vitro study of HeLa cells, UP was observed to be internalized in the cells and survived there at least 14 days [64]. UP is detected by several Toll-like receptors (TLRs), including TLRs 2 and 9, which are both expressed at the apical surface membrane of enterocytes [65,66] and upregulated in the fetal ovine gut following IA UP exposure [67]. At the upper crypts of the colon, sentinel goblet cells are present [68]. Upon TLR-MyD88 signaling, sentinel goblet cells massively release MUC2 and stimulate MUC2 release from adjacent goblet cells through NLRP6 inflammasome formation; subsequently, sentinel goblet cells are expelled into the mucus [68]. It is tempting to speculate that chronic microbial exposure, as during ongoing intra-uterine infection, could result in exhaustion of this compensatory mechanism and subsequent generation of mucus of a poor quality, as recently suggested for patients with ulcerative colitis [23,25]. Alternatively, luminal processes such as pH changes due to intra-uterine infection with UP could contribute to loss of mucus barrier functional integrity following secretion from goblet cells. UP thrives at pH levels between 6 and 7 and can increase local pH by converting urea to CO_2_ and ammonia by its urease enzyme [69,70]. In an ovine chorioamnionitis model, chronic IA infection (69 d) with UP was observed to significantly increase amniotic fluid and lung fluid pH [71]. Although in this study, 7 d IA UP exposure did not have a statistically significant effect on either amniotic fluid or lung fluid pH, it is probable that at this time point UP could cause pH disturbances in its direct environment. Mucus packing and secretion are two pH-dependent processes essential for preserving the functional integrity of the colonic mucus layer [72,73] that could be disturbed by UP. Finally, ammonia produced by UP could directly harm the colonic mucus barrier, as it was previously shown to cause intestinal injury and inflammation with a decrease in intestinal MUC2 expression in broiler chickens [74] and induced oxidative stress and mitochondrial dysfunction in Caco-2 cells [75].

Our TEM findings of cellular injury and mitochondrial disruptions in the proximal colon of IA UP-exposed lambs were concordant with the findings in the colons of NEC patients, underlining the clinical relevance of these pre-clinical observations. Although, based on conventional histological and immunohistochemical staining, only mild injury was observed following UP exposure, compelling evidence for severe and unexpected injury was seen at the cellular level with TEM. Our data emphasize that UP is not just an innocent bystander but seems to contribute to significant gastrointestinal damage already prenatally. These in utero effects may lead to a hampered mucus barrier in premature neonates. It is tempting to speculate that in the context of rapid postnatal microbial colonization of the gut, mucus barrier disruption at birth could contribute to postnatal dysbiosis, increased vulnerability to additional pro-inflammatory hits such as mechanical ventilation and sepsis [76,77] and thereby an increased risk of subsequent NEC development. Moreover, the observed disease processes (ER stress and oxidative stress) could, if persisting after birth, also induce NEC development through mucus-independent mechanisms, such as induction of intestinal inflammation and epithelial cell death.

Since ER stress and mitochondrial dysfunction/oxidative stress are causally linked in this study to colonic mucus barrier disruptions following antenatal UP exposure, treatments targeting these mechanisms may be of value to improve gut health and protect against NEC development. In line with this, in a recent study, pharmacological reduction of ER stress with tauroursodeoxycholic acid in wild type mice and murine ex-vivo explants led to increased mucus layer thickness and increased mucus growth rate with sustained mucus quality [78]. Feeding interventions are a promising strategy for addressing the pathophysiological mechanisms involved, as they are low invasive, are arguably safe and they show beneficial effects on a broad range of pathophysiological mechanisms involved in NEC, including ER stress and oxidative stress [79]. For instance, preventive enteral administration of fish oil rich in docosahexaenoic acid and eicosapentaenoic acid reduced ER stress and NEC severity in a rat NEC model [80] and treatment with milk fat globule membrane prevented a reduction in the antioxidant enzyme superoxide dismutase in a rat NEC model [81]. Since NEC only develops after an apparent disease-free interval [82], there is a postnatal window of opportunity to initiate such preventive therapies targeting ongoing disease processes, such as ER stress and oxidative stress. In addition, prenatal therapeutic strategies could be valuable in the future [13]. Of note, in a recent preclinical proof-of-concept study, we demonstrated the feasibility of screening for chorioamnionitis using volatile organic compound (VOC) detection [83].

The ovine chorioamnionitis model provides valuable and clinically relevant insight into the pathophysiological mechanisms of chorioamnionitis and related gut alterations. Using this model in combination with novel mucus integrity measurement techniques enabled us, for the first time, to assess antenatal mucus barrier integrity. These findings, along with clinical NEC samples, contribute to our understanding of chorioamnionitis pathophysiology and its relation to NEC development. An inherent limitation of the model is the limited number of animals per experimental group. In addition, we were only able to study mucus barrier changes at one time point. A study with multiple time points following intra-uterine UP infection would enable us to investigate changes in goblet cell function and mucus quality over time and to examine pathophysiological mechanisms in more detail. In addition, to gain more insight into the behavior of intestinal goblet cells and colonocytes, cell culture models, such as colon organoids, should be used to further unravel the dynamic effects of UP exposure on mucus secretion processes. Finally, future studies should be performed to shed more light on the postnatal functional consequences of the observed in utero changes and to further expand our understanding of the postulated disease mechanisms involved.

## 4. Materials and Methods

### 4.1. Experimental Design and Tissue Sampling

The current study design and ovine model have been previously described and published [53]. A translational sheep model was used to study the effects of UP-induced chorioamnionitis on fetal gut outcomes, as the gestational duration and development of the ovine intestinal tract closely resemble the human situation. This allows for a more precise timing of prenatal pro-inflammatory events, such as intra-amniotic exposure to UP. Twin fetuses from ten time-mated Texel ewes were randomly assigned to two treatment groups consisting of eight (UP) and ten (control) animals. Both amniotic sacs of the ewes were IA injected under ultrasound guidance either with saline or UP serovar 3 (HPA5 strain, 107 color changing units (CCU)) 7 days before preterm delivery at 129 days of GA (~150 d GA is normal gestational term) (Figure 11). Animals were group-housed in the animal facility from 2 days prior to the start of the experiment onwards (2-day acclimatization period). During the 7 day study period, ewes were closely monitored with a welfare diary with emphasis on inspection of the injection site and appetite and regular checks of vital parameters. Humane endpoints were pending labor, intra-uterine fetal death, and local inflammation at the injection site or signs of systemic infection not responding to treatment. Two days before delivery, all ewes received an intramuscular injection of dexamethasone 6 mg intramuscular to mimic clinical corticosteroid administration during imminent preterm birth. The twin fetuses were preterm delivered via cesarean section and immediately euthanized (at 129 d GA, which is comparable to 30–32 weeks of human intestinal development) by intravenous injection of sodium-pentobarbital (1 g). One control lamb was excluded from all analyses because it reached a humane endpoint before completion of the study. In addition, one control lamb was excluded from IHC analysis with 4% paraformaldehyde (PFA) fixed tissue, mucus thickness measurement, and qPCR and GSH/GSSG measurements due to technical problems with tissue preservation. For the mucus integrity measurements, 3 control animals and 4 UP-exposed animals were included; tissue from the other animals was used for assay optimization. Finally, for TEM analysis, 5 control animals and 6 UP animals were imaged.

Proximal colon tissue was freshly sampled and directly stored in ice cold Krebs buffer (NaHCO_3_ 24.9 mM, KH_2_PO_4_ 1.2 mM, MgSO_4_·7H_2_O 1.1 mM, KCl 4.7 mM, NaCl 118.2 mM and CaCl·2H_2_O 2.5 mM) to measure the mucus thickness and mucus barrier integrity. Additional colonic samples were fixed in 4% PFA (11699408, VWR chemicals, Radnor, PA, USA) for immunohistochemical staining and Carnoy solution (60% methanol (31721.M1, Thermo Fisher Scientific, Waltham, MA, USA), 30% chloroform (22711.324, VWR chemicals) and 10% glacial acetic acid (20102292, VWR chemicals)) for both immunohistochemical and immunofluorescence staining. For TEM purposes, colon samples were fixated in Karnovsky fixative (2.5% glutaraldehyde and 2% paraformaldehyde in sodium cacodylate 0.1 M; pH 7.4) and stored in storage buffer (2% paraformaldehyde in phosphate buffer (PB)) at 4 °C. Lastly, samples were also collected in liquid nitrogen for performing the glutathione (GSH/GSSG) assay and real-time quantitative polymerase chain reactions (RT-qPCRs).

### 4.2. Mucus Thickness Measurement and Mucus Barrier Functional Integrity Measurement

Colonic samples were stored in Krebs buffer and mucus thickness/mucus barrier function integrity was measured within three hours after the sampling procedure. To study the effects of intra-uterine UP exposure on the colonic mucus barrier of preterm lambs, mucus thickness measurements were performed as a functional parameter for mucus secretion by goblet cells. The colonic mucus thickness measurement technique was based on the method described by Gustafsson et al. [56] and applied to premature ileal samples in our ovine chorioamnionitis model [53]. Briefly, proximal colon samples were cut open and gently macerated in Krebs buffer to remove stool; no flushing was performed. Thereafter, colon samples were stretched on a silicone-coated Petri dish with the crypt side upwards. Subsequently, 10 µm black dyed microspheres (24294-2, Polysciences, Warrington, PA, USA) were added to the apical side of the tissue to visualize and measure the TML thickness by adding contrast, as the mucus layer is transparent and difficult to measure and fixation of tissue leads to mucus loss or collapse [19,56]. The colonic TML thickness (inner plus outer mucus layer) was assessed by measuring the distance between the 10 µm mucus-attached beads and the colonic epithelium using a home-made microneedle (with a ~25 µm tip diameter) connected to a custom-made micromanipulator (Debets Mechanical Support, Stein, The Netherlands) at a 45° angle (Figure 12A). The vertical mucus thickness was calculated by multiplying the 45° bead-epithelium distance by cosine 45 [53,56]. Five different regions were measured, and the median was calculated to determine the total mucus thickness value (µm) per animal.

For measuring the mucus functional integrity, 1 µm red-fluorescent beads (Fluospheres, 11584766, Thermo Fisher Scientific, Waltham, MA, USA) were used to assess the integrity function of the colonic IML [56]. Adjacent colonic tissue samples were used from those used for mucus thickness measurement. The 1 µm polystyrene microspheres, representative of bacterial size, were dissolved in Krebs solution and added to the apical side of the stretched colonic sample and incubated for 5 min (Figure 12B). The distribution of the 1 µm beads in the colonic IML was analyzed by creating Z-stacks with a 5 µm interval using a two-photon microscope (Light Microscope Leica STP6000 2-photon, Leica Microsystems). Five different regions per animal were analyzed by calculating the distances between individual 1 µm beads and the colonic epithelium (Figure 12B). The colonic epithelium was visualized by adding a Hoechst solution (Hoechst 34580, Sigma-Aldrich, Saint Louis, MO, USA; 1 µg/mL in Krebs solution) to the apical side of the colonic sample. The median of the IML thickness (µm) was calculated for each animal and compared to the TML to assess alterations in mucus composition after IA saline/UP exposure for 7 d.

### 4.3. Immunohistochemistry

Colonic samples fixed in 4% PFA (stained for SPDEF, BiP and CHOP) and Carnoy solution (stained for ‘immature’ (non-O-glycosylated) apomucin-2 (iMUC2) and MUC2) were used for immunohistochemical (IHC) visualizations. SPDEF was used to assess goblet cell differentiation, while iMUC2 and MUC2 served as markers for intestinal goblet cells. Additionally, BiP and CHOP staining were performed to study epithelial ER stress in the preterm colon. After fixation, the colonic samples were infiltrated and embedded with paraffin and then cut into 4 µm thick sections using a microtome (RM2235, Leica Microsystems). Tissue sections were deparaffinized and rehydrated prior to blocking endogenous peroxidase activity (0.3% H_2_O_2_ diluted in phosphate buffered saline (PBS)). Antigen retrieval was performed by boiling the colonic sections in a sodium-citrate buffer (10 mM, pH 6.0) for 10 min. Subsequently, non-specific binding sites were blocked by incubating the colonic sections with 5% bovine serum albumin (BSA) in PBS for 30 min. Thereafter, the colonic sections were incubated with the primary antibody of interest (Appendix A). Sections were washed and incubated with the appropriate biotin-conjugated secondary antibody (Appendix A) and the signal was subsequently enhanced by incubating the sections in an avidin-biotin complex kit (Vectstain Elite ABC kit, PK6100, Maravai LifeSciences, San Diego, CA, USA). Visualization of the immunoreaction for all the immunohistochemical staining was performed by incubation of the section in 3,3′-diaminobenzidine tetrahydrochloride (DAB, Thermo Fisher Scientific, Waltham, MA, USA). No counterstaining was applied for all the immunohistochemical staining (MUC2, iMUC2, SPDEF, BiP and CHOP).

### 4.4. Analysis of IHC-Stained Slides

IHC-stained slides were scanned with the use of the Ventana Iscan (Ventana Medical System, INc., Tucson, AZ, USA) at a magnification of 400×. Positively stained cells were counted (semi-automated fashion) for MUC2, iMUC2, SPDEF and CHOP IHC-reactivity in two colonic cross sections across the whole colonic crypts using QuPath software (v0.2.3, University of Edinburgh, Edinburgh, UK) [84]. The positive cell count was corrected for the colonic tissue surface (measured in QuPath) and represented as positive cells per mm^2^ of the colonic tissue surface. The average cell count per sheep is presented, and analyses were performed blinded by one researcher. The intensity of the BiP staining was assessed blinded by two independent investigators according to a scoring system previously applied and published [53,85]: 0: negatively stained; 1: mild/slight staining; 2: moderate staining; 3: intense/strong staining.

### 4.5. GSH and GSSG Measurement

The ratio between reduced glutathione (GSH) and its oxidized product glutathione disulphide (GSSG) was used as a marker for cellular redox status [38]. Proximal colonic samples were frozen in liquid nitrogen and stored at −80 °C until use for determining the concentrations of GSH, GSSG and the ratio of GSH:GSSG in colon samples. A piece of colon (weight ranging from 52.3–76 mg) was defrosted and the meconium was removed from the sample. Subsequently, the sample was lysed/homogenized and normalized to a concentration of 250 mg/mL in lysis buffer (0.1 M Potassium Phosphate buffer containing 10 mM EDTA disodium salt, pH 7.5 and 1% Triton-X-100) for 30 min on ice. Samples were centrifuged for 10 min at 14,000 rpm and 4 °C, and part of the supernatant was used to measure the protein content using the bicinchoninic acid assay (BCA assay; 23225, Thermo Fisher Scientific). The remaining supernatant was diluted 1:1 in 6% sulfosalicylic acid. Finally, 0.1 M lysis buffer without Triton-X-100 was added to this solution in a ratio of 1:10 and GSH and GSSG concentrations in the proximal colon were determined using the enzymatic recycling method as described previously [86]. The absorbance was measured for 10 min at 412 nm at 37 °C using a microplate reader (Synergy HTX Multi-Mode Reader, Agilent Technologies, Santa Clara, CA, USA). GSH and GSSG concentrations were calculated based on the slope of the absorbance curve and corrected for the total protein concentration. The final data are represented in nmol/mg protein. These values were also used to calculate the GSH:GSSG ratio of the preterm ovine colonic samples.

### 4.6. RNA Isolation, cDNA Synthesis and RT-qPCR

RNA isolation, cDNA synthesis and RT-qPCR were performed as previously described [85]. Briefly, following lysation and homogenization of snap-frozen colon samples, RNA was isolated using the RNeasy Mini Kit (#74104, Qiagen, Hilden, Germany). cDNA synthesis was performed using the SensiFast cDNA synthesis kit (BIO-65054, Bioline, London, UK), and real-time (RT) qPCR was performed with the SensiMix SYBR Hi-Rox kit (QT605–05, Bioline) and LightCycler-580 Instrument (Roche Applied Science, Basel, Switzerland). The ovine-specific primers used are summarized in Appendix A. To assess the cellular redox state, the mRNA expression of three Nrf2-mediated genes was analyzed [87]. Glutamate-cysteine ligase catalytic subunit (GCLC) is the catalytic subunit of the enzyme glutamate cysteine ligase that catalyzes the rate-limiting step in the cellular formation of the antioxidant GSH. Thioredoxin reductase 1 (TXNRD1) is part of the thioredoxin system that is essential for cellular redox homeostasis; TXNRD1 recycles oxidized TXN to its reduced form, which can be used again for reducing ROS [88]. Finally, heme oxygenase 1 (HMOX1) is the rate-limiting enzyme in heme degradation and is considered a strong antioxidative agent [89,90]. The mRNA expression of GCLC, TXNRD1 and HMOX1 is regulated by various cellular stimuli, including oxidative stress [91]. LinRegPCR software (version 2016.0, Heart Failure Research Center, Academic Medical Center, Amsterdam, The Netherlands) was used for RT-qPCR data processing. Data were normalized with the geometric mean of three housekeeping genes (ovRPS15, GAPDH and YWHAZ), and fold changes compared to controls were calculated with the N0 values.

### 4.7. Human Intestinal Sample Collection

Intestinal samples were collected from three pediatric patients undergoing colonic resection for acute NEC (patients 1 and 2), stoma reversal after NEC (patient 2) and revision of colostomy after partial colon resection for colorectal malformation (patient 3). Patients 1 and 2 were born preterm and treated at the neonatal intensive care unit of Maastricht University Medical Centre. Patient 3 was born at term and treated at the pediatric ward of Maastricht University Medical Centre. After tissue collection, the colon was kept ice-cold and fixed using Karnovsky fixative (2.5% glutaraldehyde and 2% paraformaldehyde in sodium cacodylate 0.1 M; pH 7.4) and processed as outlined for the ovine colonic samples (described in section below).

### 4.8. Transmission Electron Microscopy (TEM)

Karnovsky fixed colonic samples were further processed by post-fixation at 4 °C with 1% osmium tetroxide in sodium cacodylate (0.1 M) containing potassium ferrocyanide (0.8%) and subsequently dehydrated with ethanol. In the final step, fixed colonic tissue was infiltrated, embedded and polymerized into Epon resin for at least 48 h at 60 °C. Ultrathin sections (70 nm thick) were cut using an ultramicrotome (Ultracut UC6, Leica Microsystems). For TEM imaging, saline (N = 4) and UP-exposed (N = 6) colonic samples were randomly selected and imaged. For human samples, control (N = 2) and NEC (N = 2) colonic samples were used. The ultrathin-cut sections were mounted on Formvar-coated copper grids (single slot grids) and subsequently stained with 2% uranyl acetate in 50% ethanol and lead citrate. The TEM-prepared colonic samples were imaged using a Tecnai G2 Spirit transmission electron microscope with a CCD Eagle camera (4 k × 4 k camera; Thermo Fisher Scientific).

### 4.9. Immunofluorescence Staining of Ureaplasma parvum

Carnoy-fixed colonic samples were used to perform immunofluorescence staining to detect UP serovar 3 in the proximal colon. Colonic sections with a thickness of 4 µm were deparaffinized and rehydrated prior to the immunofluorescent staining. Thereafter, the sections were washed with PBS and subsequently incubated with primary UP antibody (GEN313159, Gentaur Europe, Kampenhout, Belgium) for 1 h. After washing with PBS, the slides were incubated with a fluorescent-labeled secondary antibody (A-31570, polyclonal donkey anti-mouse Alexa 555; Thermo Fisher Scientific) for 60 min. To visualize the colonic epithelium, the slides were stained with 4′,6-diamidino-2-phenylindole (DAPI; 200 µg/mL D9542, Sigma Aldrich, St. Louis, MO, USA). Sections were embedded using fluorescent mounting medium (53023, Agilent Dako, Glostrup, Denmark). UP-positive cells were imaged with confocal microscopy (LEICA DMI 4000 confocal microscope (Leica Microsystems, Wetzlar, Germany) using the 63× oil objective.

### 4.10. Ethics

Animal experiments were approved by the animal ethics committee of Maastricht University (Maastricht, The Netherlands; registration number: PV2015-005-002). Human sample collection was approved by the Medical Ethical Committee of Maastricht University Medical Centre (registration number 1–5–185). Written informed consent was given by the parents of all three patients (one patient twice). All authors had access to the study data and reviewed and approved the final manuscript.

### 4.11. Statistics

GraphPad Prism Software (v6.0, Graphapd software Inc., La Jolla, CA, USA) was used for statistical analyses. The statistical significance between the two different treatment groups was tested using the Mann–Whitney U test. A difference was interpreted as statistically significant in the case of *p* < 0.05. Since the number of animals per group was relatively low, *p*-values between 0.05 and 0.10 are reported in exact numbers, as previously described [13,53,85]. This approach reduces the risk of missing potentially biologically relevant findings but increases the chance of false negative findings. For all data graphs, data are represented as the median and interquartile range (IQR).

## 5. Conclusions

In conclusion, UP-induced chorioamnionitis leads to a dysfunctional colonic mucus barrier and goblet cell hypersecretion, which is paralleled with mucosal cell alterations, including ER stress and mitochondrial dysfunction. This emphasizes the pathogenicity of UP in this context. In addition, the cellular alterations largely overlap with findings in colonic human NEC biopsies, providing a strong mechanistic link between chorioamnionitis and NEC pathophysiology. These adverse outcomes already observed in utero following chorioamnionitis could predispose vulnerable children to additional postnatal pro-inflammatory insults and increase the risk for NEC development later in life.

## Figures and Tables

**Figure 1 ijms-25-04000-f001:**
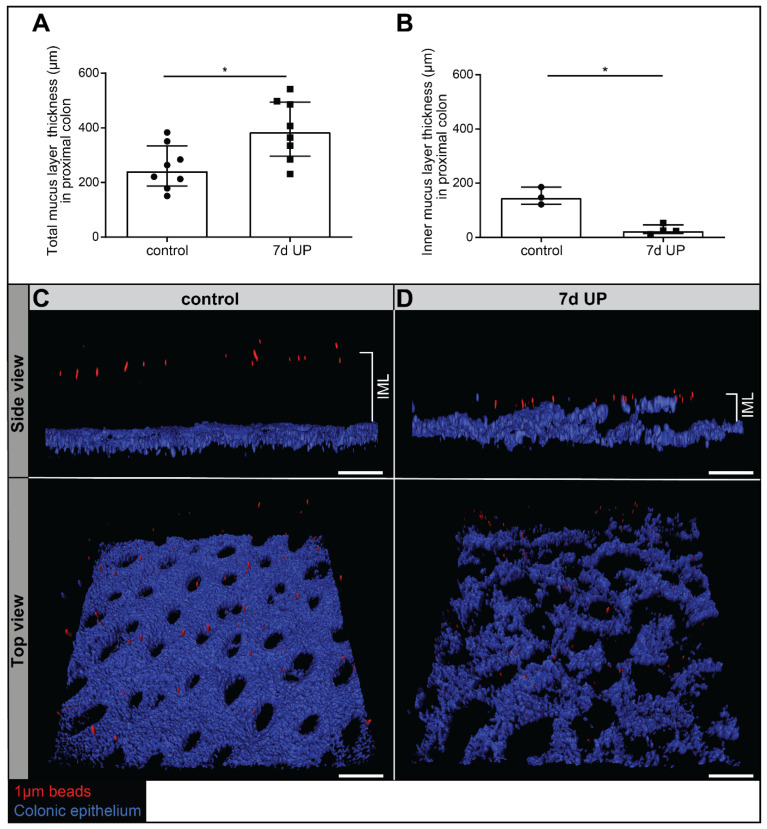
TML thickness and mucus barrier functional integrity (IML thickness) in the premature ovine colon. Total mucus thickness, evaluated ex vivo with 10 µm beads and a custom-made mucus measurement system, was statistically significantly increased in 7 d IA UP-exposed animals compared to preterm controls ((**A**), control N = 8, UP N = 8). Mucus barrier functional integrity was measured ex vivo in control and 7 d IA UP-exposed lambs with 1 µm (bacterial sized) beads and two photon microscopy. Compared to premature control lambs ((**B**,**C**) N = 3), mucus barrier functional integrity (IML thickness) was decreased in 7 d IA UP-exposed animals ((**B**,**D**), N = 4), with increased penetration of the 1 µm beads (red) towards the colonic epithelium (blue). In UP-exposed lambs, numerous beads reached the intestinal epithelium (**B**,**D**). In addition, whereas the organization of the colonic epithelium was intact in control premature lambs (**C**), this was disrupted following IA UP exposure (**D**). Data are displayed as the median ± interquartile range. Scale bars indicate 100 µm. * *p* < 0.05. Abbreviations: IA: intra-amniotic; IML: inner mucus layer; UP: *Ureaplasma parvum*.

**Figure 2 ijms-25-04000-f002:**
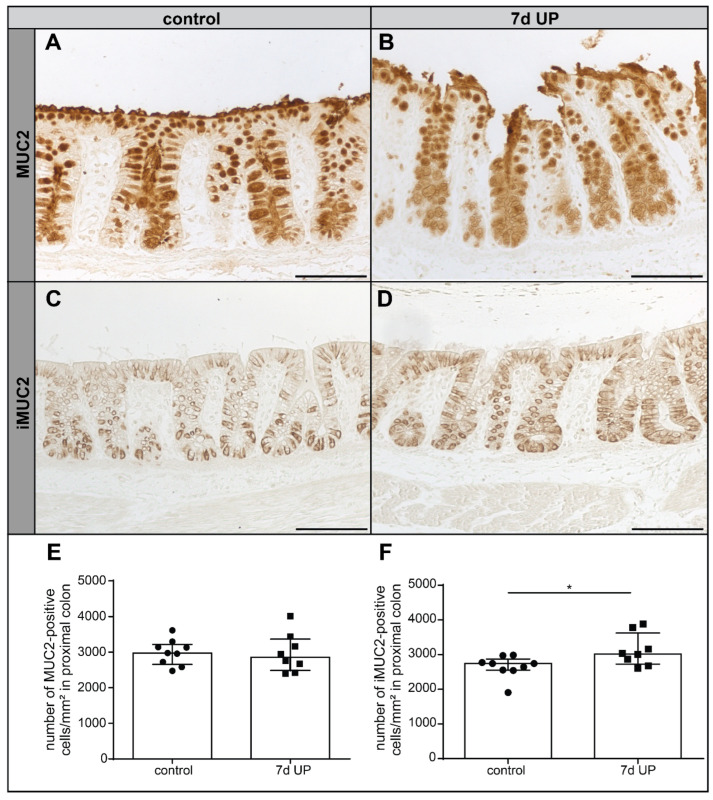
Number of MUC2+ and iMUC2+ goblet cells in the proximal colon of premature lambs per mm^2^ tissue surface area. No changes in the number of MUC2+ goblet cells were observed between controls ((**A**,**E**), N = 9) and 7 d IA UP-exposed animals ((**B**,**E**), N = 8). Compared to controls ((**C**,**F**), N = 9), the number of iMUC2+ cells was statistically significantly increased in IA UP-exposed animals ((**D**,**F**), N = 8) and a more intense staining of iMUC+ goblet cells was observed in the upper crypt of 7 d IA UP-exposed animals (**D**) than in the upper crypt of controls (**C**). Each data point represents the average positive cell count of one animal of the MUC2 (**E**) and iMUC2 (**F**) immunohistochemical staining, respectively. Data are displayed as the median with the interquartile range. Scale bars indicate 100 µm. * *p* < 0.05. Abbreviations: IA: intra-amniotic; iMUC2: ‘immature’ apomucin-2; MUC2: mucin-2; UP: *Ureaplasma parvum*.

**Figure 3 ijms-25-04000-f003:**
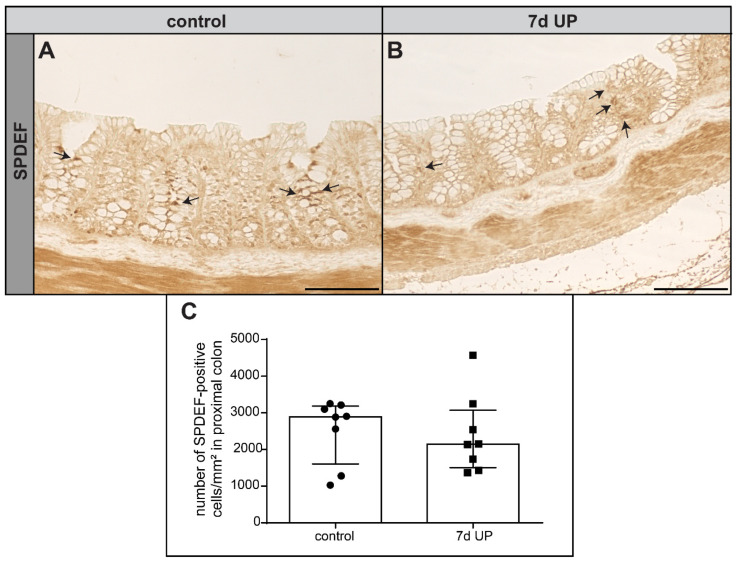
Number of SPDEF+ goblet cells in the proximal colon of premature lambs per mm^2^ tissue surface area. The number of SPDEF+ cells (black arrows) was unaltered compared to control premature lambs ((**A**,**C**), N = 8) following IA UP exposure ((**B**,**C**), N = 8). Each data point represents the average positive cell count of SPDEF immunohistochemical staining. Data are displayed as the median with the interquartile range. Scale bars indicate 100 µm. Abbreviations: IA: intra-amniotic; SPDEF: Sterile Alpha Motif Pointed Domain-containing Ets Transcription Factor; UP: *Ureaplasma parvum*.

**Figure 4 ijms-25-04000-f004:**
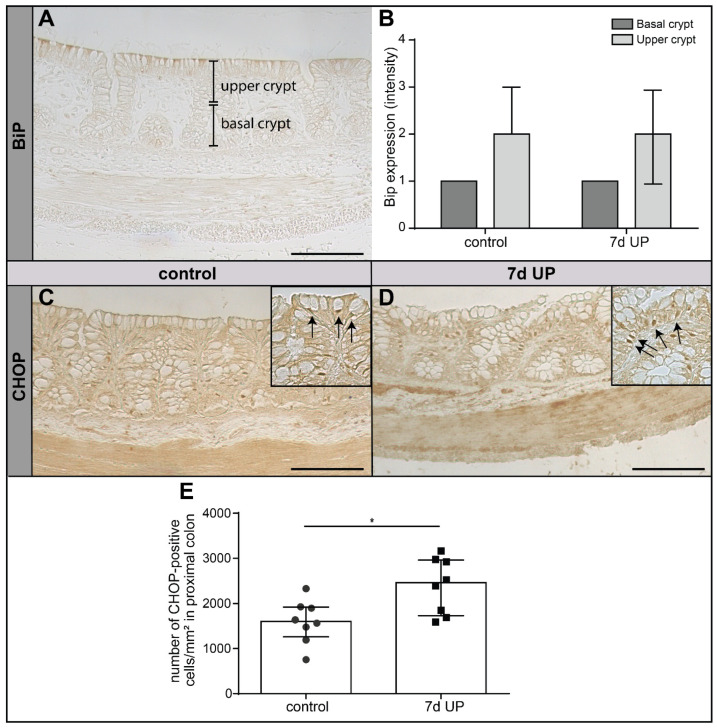
BiP expression levels and the number of CHOP+ epithelial cells (black arrows) per mm^2^ tissue surface area in the proximal colons of premature lambs. BiP expression in the upper crypt was higher compared to the lower crypt in both the IA UP-exposed and control lambs ((**A**,**B**), control N = 8; UP N = 8). No difference was observed in BiP expression levels between the groups ((**B**), control N = 8; UP N = 8). Compared to controls ((**C**,**E**), N = 8), 7 d IA UP exposure leads to a statistically significant increase in CHOP+ epithelial cells ((**D**,**E**), N = 8). Each data point represents the average positive cell count of one animal in the CHOP immunohistochemical staining (**E**). Data are displayed as the median with the interquartile ranges. Scale bars indicate 100 µm. * *p* < 0.05. Abbreviations: IA: intra-amniotic; BiP: binding immunoglobulin protein; CHOP: C/EBP homologous protein; UP: *Ureaplasma parvum*.

**Figure 5 ijms-25-04000-f005:**
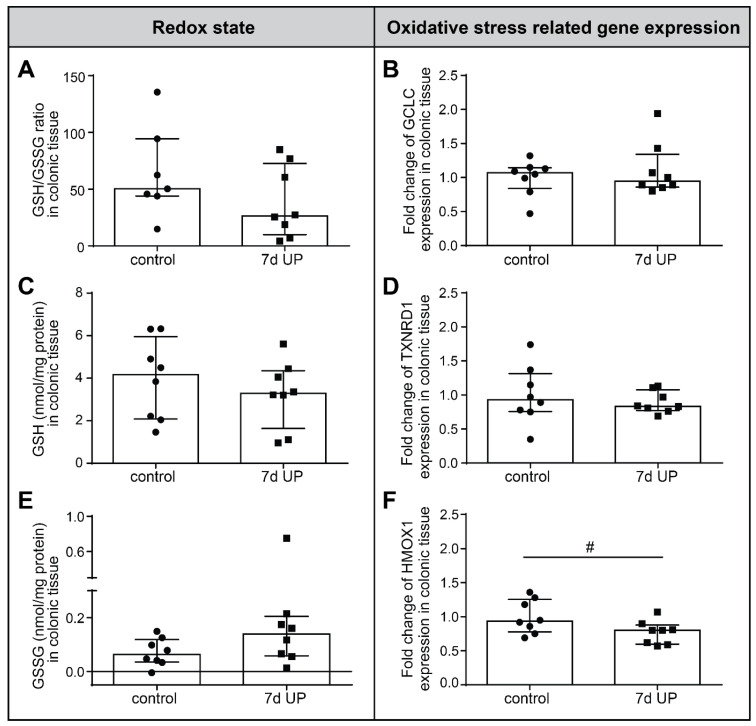
Redox state (GSH/GSSG ratio) and mRNA expression of oxidative stress in the proximal colon of preterm lambs. GSH/GSSG ratio ((**A**), control N = 7; UP N = 8), determined by dividing GSH protein expression ((**C**), control N = 8; UP N = 8) by GSSG protein expression ((**E**), control N = 8; UP N = 8) is lower, albeit not statistically significant, following 7 d IA UP exposure compared to controls; 5 out of 8 UP exposed animals have a GSH/GSSG ratio below the interquartile range of controls compared to 1 out of 7 control animals. mRNA expression of GCLC ((**B**), control N = 8; UP N = 8) and TXNRD1 ((**D**), control N = 8; UP N = 8) was not altered between the control and 7 d UP-exposed group. Interestingly, the gene expression of HMOX1 tended to be lower in 7 d UP-exposed animals compared to controls ((**F**), control N = 8; UP N = 8). Each data point represents the GSH/GSSG protein expression ratio (**A**), average protein concentration (GSH, GSSG; (**C**,**E**)) or relative mRNA expression (GCLC, TXNRD1 and HMOX1; (**B**,**D**,**F**)) of one lamb. Data are displayed as the median with the interquartile range. # 0.05 < *p* < 0.1. Abbreviations: GCLC: glutamate-cysteine ligase catalytic subunit; GSH: glutathione; GSSG: oxidized glutathione; HMOX1: heme oxygenase-1; IA: intra-amniotic; Redox: reduction-oxidation; TXNRD1: thioredoxin reductase 1; UP: *Ureaplasma parvum*.

**Figure 6 ijms-25-04000-f006:**
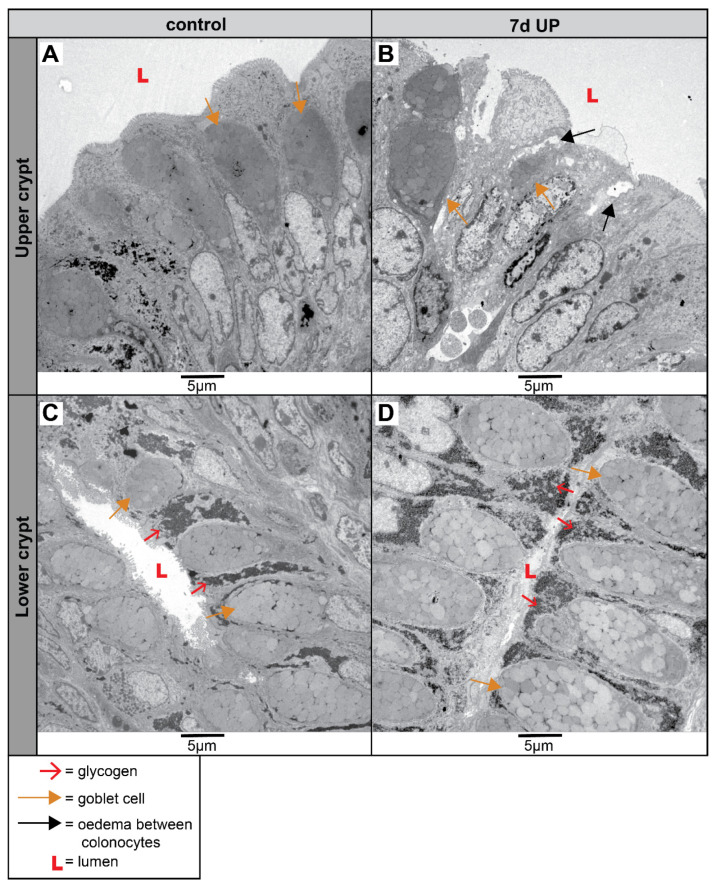
Cellular morphology of colonocytes and goblet cells and tissue organization in the upper and lower crypts of the proximal colons of preterm lambs imaged with TEM. 7 d IA UP exposure leads to damage of the colonic epithelial layer at the upper colonic crypt, characterized by edema (black arrows) between the mucosal barrier cells; this was not seen in control lambs ((**A**,**B**), control N = 5; UP N = 6). Compared to controls (**C**), an accumulation of glycogen (red arrowhead) was observed both in colonocytes and goblet cells in the lower crypt in 7 d IA UP-exposed lambs (**D**). The lumen is shown by L. Scale bars indicate 5 µm. Abbreviations: IA: intra-amniotic, TEM: transmission electron microscopy, UP: *Ureaplasma parvum*.

**Figure 7 ijms-25-04000-f007:**
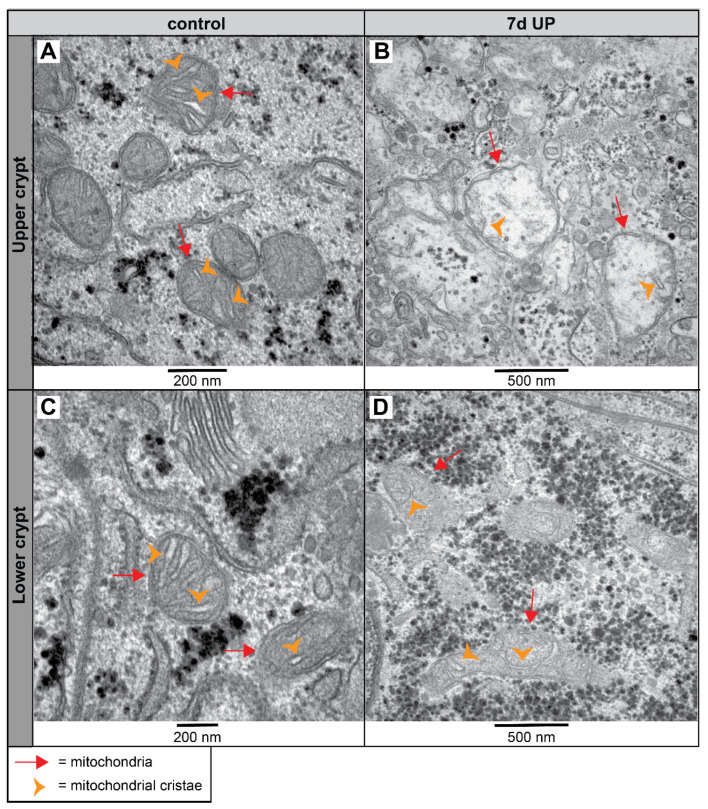
Mitochondrial morphological alterations in the colonocytes of the lower and upper crypt of the proximal colon in preterm lambs visualized with TEM imaging. Compared to parallelized structured cristae (orange arrowhead) of the mitochondria (red arrow) of premature controls ((**A**), N = 5), the mitochondria (red arrow) in the upper crypt of 7 d IA UP-exposed animals have a globular shape with disrupted and fewer organized cristae (orange arrowhead) ((**B**), N = 6). Compared to controls ((**C**), N = 5), mitochondria (red arrow) in the lower crypts of 7 d UP-exposed lambs appeared to have lobular-shaped (C-shape) cristae ((**D**), N = 6). Scale bars indicate 200 nm and 500 nm, respectively. Abbreviations: IA: intra-amniotic; TEM: transmission electron microscopy; UP: *Ureaplasma parvum*.

**Figure 8 ijms-25-04000-f008:**
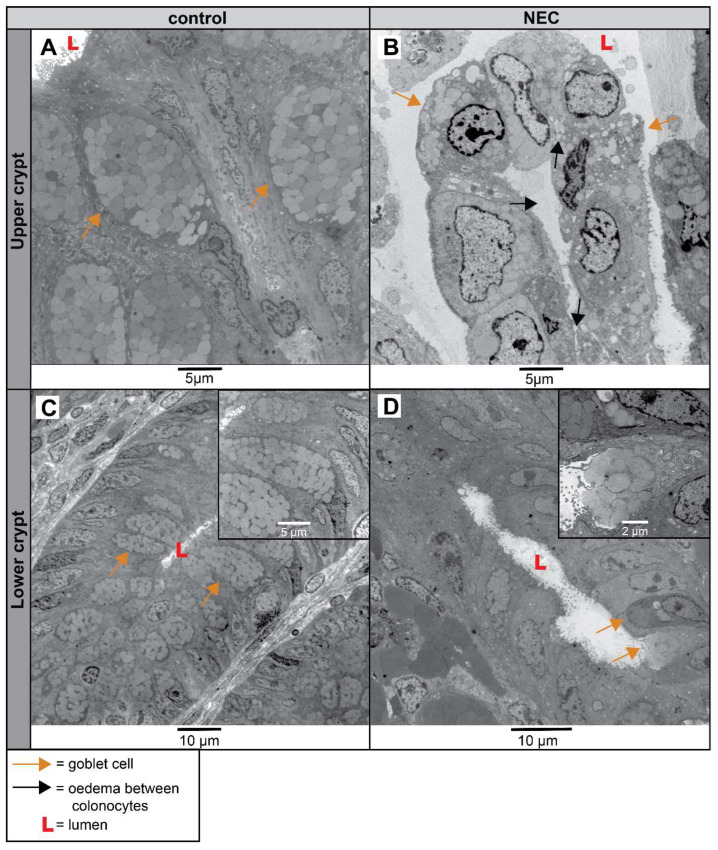
Cellular morphology and tissue organization in the upper and lower colonic crypts of NEC patients and controls visualized with TEM. In the upper crypt of the controls, an abundant amount of goblet cells (orange arrows) with a recognizable structure and organization was observed ((**A**), N = 2). In contrast, the organization of the colonic upper crypt of NEC patients was severely disrupted; no clear cell types (colonocytes or goblet cells) can be identified based on morphology, and edema (black arrows) was seen between cells ((**B**), N = 2). The cellular morphology and organization of the goblet cells (orange arrows) and colonocytes present in the lower crypt are normal in control infants ((**C**)), N = 2), whereas disorganization and reduced numbers of intact goblet cells (orange arrows and insert) were observed in the lower crypts of NEC patients ((**D**), N = 2). Scale bars indicate 5 µm and 10 µm respectively. Abbreviations: L: lumen; NEC: necrotizing enterocolitis; TEM: transmission electron microscopy.

**Figure 9 ijms-25-04000-f009:**
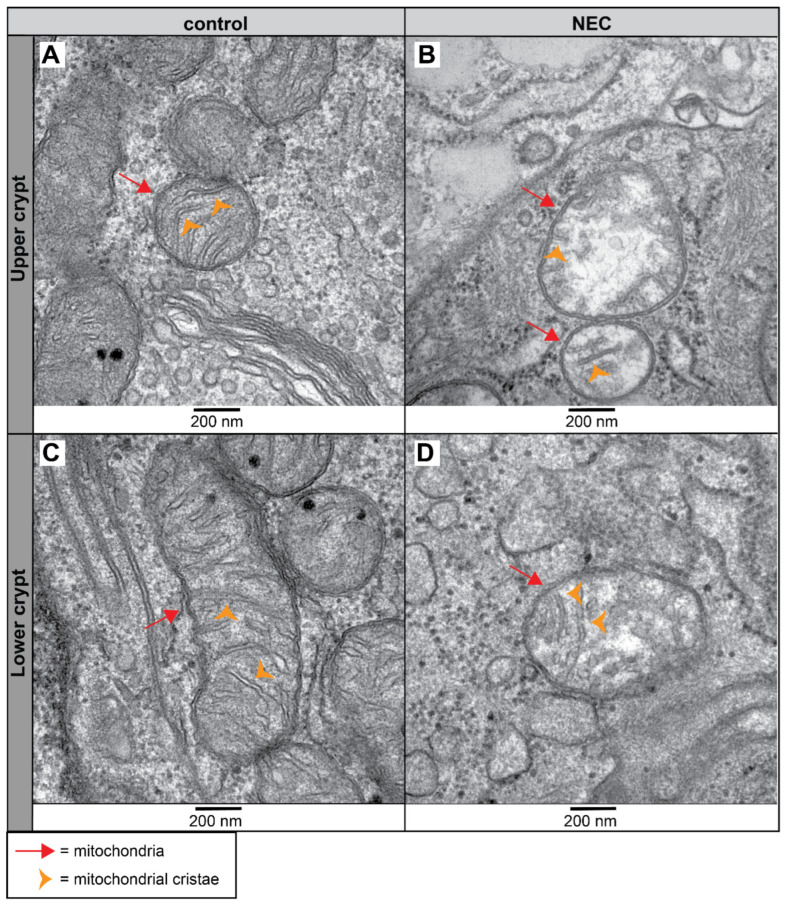
Morphological alterations of the mitochondria in the colonocytes of the upper and lower crypts of NEC patients and controls imaged with TEM. Compared to the normal morphological appearance of mitochondria (red arrow) in the upper crypt of controls ((**A**), N = 2), the mitochondria (red arrow) in the upper crypt of NEC patients are disorganized with disturbed cristae (orange arrowhead) ((**B**), N = 2). Mitochondria (red arrow) in the lower crypts of the controls show an elongated morphology with an abundance of parallel organized cristae (orange arrowhead) ((**C**), N = 2). In NEC patients, the mitochondrial morphology is altered in the lower crypt; mitochondria (red arrow) are more globular shaped, containing a less electron dense matrix and have a lower number and disrupted cristae (orange arrowhead) ((**D**), N = 2). Scale bars indicate 200 nm. Abbreviations: NEC: necrotizing enterocolitis; TEM: transmission electron microscopy.

**Figure 10 ijms-25-04000-f010:**
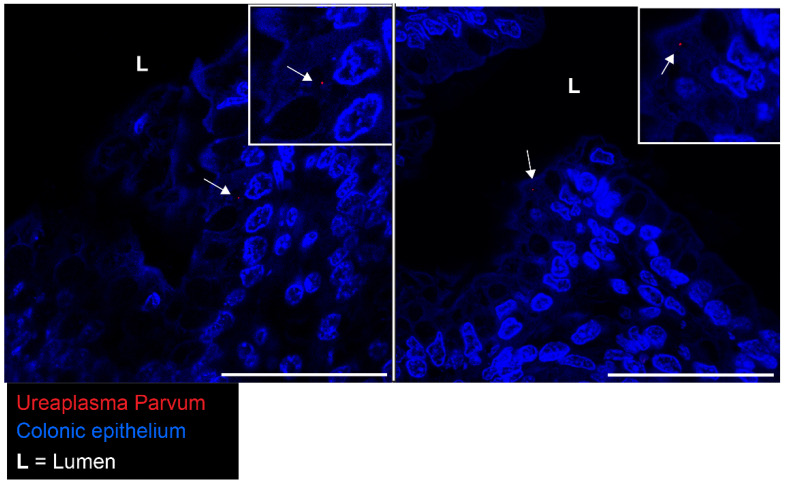
Detection of UP in proximal colonic tissue of premature lambs with IF staining. UP (indicated by white arrows) is present in the epithelial cells of premature lambs IA exposed to UP for 7 days. Two examples are shown. Control N = 9, UP N = 8. Scale bars indicate 50 µm. Abbreviations: IA: intra-amniotic; IF: immunofluorescence; L: lumen; UP: *Ureaplasma parvum*.

**Figure 11 ijms-25-04000-f011:**
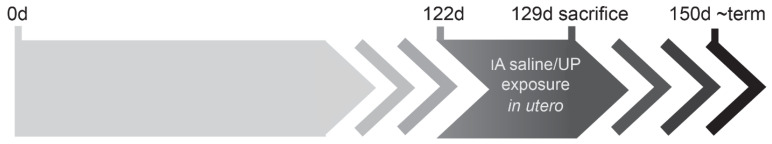
Study design of our ovine UP-induced chorioamnionitis model. Twin fetuses from time-mated Texel ewes were randomly assigned to two different study groups. Twins (both amniotic sacs) were IA injected with saline or UP serovar 3 (107 CCU) under ultrasound guidance at 122 d of GA, and IA injections were given 7 d before preterm birth (via cesarean section) at 129 d of GA (term~150 d). Abbreviations: CCU: color-changing units; GA: gestational age; IA: intra-amniotic; UP: *Ureaplasma parvum*.

**Figure 12 ijms-25-04000-f012:**
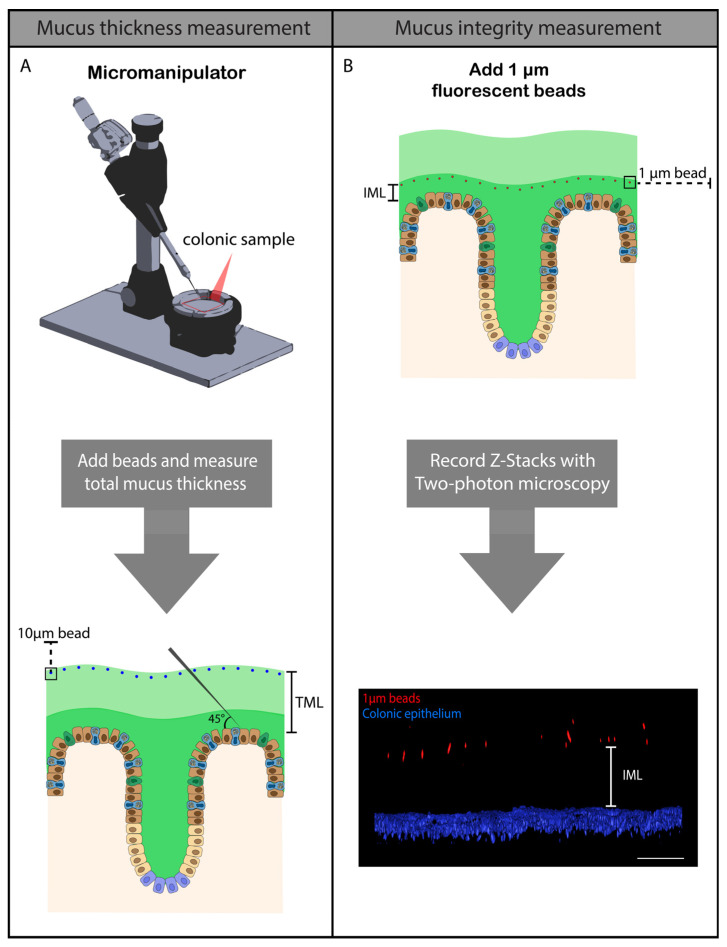
Methodology of mucus thickness measurement and mucus integrity measurement of proximal preterm colon. (**A**) For TML thickness measurement, colon samples were cut open, macerated in Krebs buffer and stretched on a silicone coated Petri dish. Thereafter, black dyed microspheres were added to the apical side of the tissue and TML thickness was assessed by measuring the distance between the 10 µm mucus-attached beads and the colonic epithelium using a home-made microneedle on a custom-made micromanipulator at a 45° angle. The vertical mucus thickness was calculated by multiplying the 45° bead-epithelium distance by cosine 45. Five different regions were measured and the median was calculated to create TML thickness value per animal. (**B**) For mucus functional integrity measurement, colon samples were cut open, macerated in Krebs buffer and stretched on a silicone coated Petri dish. Thereafter, Hoechst solution (for colonic epithelium visualization) and 1 µm red-fluorescent beads (for mucus integrity visualization) were added to the apical side of the tissue and the distribution of the 1 µm beads in the colonic IML was analyzed by creating Z-stacks with a 5 µm interval using a two-photon microscope. Five regions per animal were analyzed by calculating distances between individual 1 µm beads and the colonic epithelium (IML thickness); the median of the IML thickness (µm) was calculated for each animal. Scale bar indicates 100 µm. Abbreviations: TML: total mucus layer; IML: inner mucus layer.

## Data Availability

The data that support these findings of the study are available upon request from the corresponding authors.

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
