# Peer review of "Antenatal Ureaplasma Infection Causes Colonic Mucus Barrier Defects: Implications for Intestinal Pathologies"

_ijms, 2024, doi:10.3390/ijms25074000_

Round 1
Reviewer 1 Report
Comments and Suggestions for Authors
The author wrote a manuscript entitled "Antenatal Ureaplasma infection causes colonic mucus barrier 2 defects: implications for intestinal pathologies". However, the overall quality of the manuscript is very good. The idea of paper is novel in this field.
-Abstract should be designed in the paragraph
-Introduction. The concluded paragraph should be added
-The conclusion should be rewritten
-Reference can be updated id needed
-Table and Figure should be cited in the text
Comments on the Quality of English Languageminor
Reviewer 2 Report
Comments and Suggestions for Authors
The study of Ureaplasma and the changes they produce in the postnatal functional consequences of the knowledge of the mechanisms involved in the postnatal pathological mechanisms involved in Ureaplasma.
the postnatal pathological mechanisms involved. This will allow early diagnosis to avoid fatal consequences. In addition, the work is carried out with scientific rigour, the figures are excellent. I would only ask kindly that care be taken to ensure that the name of the bacterium is written in italics or italics. If the limitation was the number of animals, perhaps the laws on animal husbandry would allow an increase in the number of animals as well as the ethics committees in the handling of animals.
excellent work
Reviewer 3 Report
Comments and Suggestions for Authors
Dear Authors,
I want to congratulate you for the hard work that was put into this manuscript. It was interesting reading it, and with a strong scientific value. However, several edits and corrections are required before publication can be considered. Please consider the following:
Abstract:
· Please rewrite in abstract in a depersonalized manner. So don’t use first person like “our study”, “we studied”, “we validated”, “we demonstrate”.
· Please include some objective data when describing, for example, “UP-exposed lambs have a thicker, but dysfunctional, colonic mucus layer”. It would be easier to understand the concept if some data is also included in the abstract.
· Clearly articulate the gap in current knowledge regarding the impact of UP on colonic barrier integrity and its potential link to NEC. This establishes the study's necessity and objectives more firmly.
· Make the research objectives explicit. Clearly state the hypothesis that antenatal UP exposure affects colonic mucus barrier integrity and its potential contribution to NEC pathogenesis.
Introduction:
· I fell like the introduction is too long. Please shorten the paragraphs and be more concise, focusing on the main topic. It’s too much to go through a 6 big paragraphs introduction when reading a 30 pages study.
Materials and Methods:
· Consider providing more details on the selection criteria for the animal model and any control measures to ensure the reliability of your results.
· For each methodological approach (e.g., mucus thickness measurement, immunohistochemistry), briefly justify your choice to help readers understand their relevance and robustness.
Results:
· Very thoroughly and correctly described. Nothing to suggest here.
Discussions:
· Dive deeper into the implications of your findings, particularly how they contribute to understanding the mechanism linking chorioamnionitis and NEC.
· While you've briefly mentioned limitations related to the animal model and study design in the discussion, providing a more thorough analysis here could help contextualize your findings. Discuss both the strengths and limitations of your approach.
· Suggest specific future research directions or potential clinical interventions based on your findings. This could include recommendations for antenatal screenings or therapeutic strategies to mitigate UP's effects on the colonic mucus barrier.
Conclusions:
· Clear and concise. Nothing to suggest here.
References:
· Correctly formatted.
Good luck!
Comments on the Quality of English LanguageEnglish language requires minor revisions.
